# Study of Through Glass Via (TGV) Using Bessel Beam, Ultrashort Two-Pulses of Laser and Selective Chemical Etching

**DOI:** 10.3390/mi14091766

**Published:** 2023-09-14

**Authors:** Jonghyeok Kim, Sungil Kim, Byungjoo Kim, Jiyeon Choi, Sanghoon Ahn

**Affiliations:** 1Department of Laser & Electron Beam Technologies, Korea Institute of Machinery & Materials, 156 Gajeongbuk-Ro, Yuseong-Gu, Daejeon 34103, Republic of Korea; imj02096@kimm.re.kr (J.K.); kims24@corning.com (S.K.); byungjookim@kimm.re.kr (B.K.); jchoi@kimm.re.kr (J.C.); 2Department of Mechanical Engineering (Robot∙Manufacturing Systems), University of Science and Technology, 217 Gajeong-Ro, Yuseong-Gu, Daejeon 34113, Republic of Korea

**Keywords:** selective laser-induced etching, burst-mode laser, spatial light modulator, Bessel beam, ultrashort pulsed laser

## Abstract

Selective laser etching is a promising candidate for the mass production of glass interposers. It comprises two steps: local modification by an ultrashort-pulsed laser and chemical etching of the modified volume. According to previous studies, when an ultrashort-pulsed laser beam is irradiated on the sample, electron excitation occurs, followed by phonon vibration. In general, the electron excitation occurs for less than a few tens of picoseconds and phonon vibration occurs for more than 100 picoseconds. Thus, in order to compare the electric absorption and thermal absorption of photons in the commercial glass, we attempt to implement an additional laser pulse of 213 ps and 10 ns after the first pulse. The modified glass sample is etched with 8 mol/L KOH solution with 110 °C to verify the effect. Here, we found that the electric absorption of photons is more effective than the thermal absorption of them. We can claim that this result helps to enhance the process speed of TGV generation.

## 1. Introduction

Various IoT devices have been developed since the introduction of the Internet of Things (IoT) by Ashton in 1999 [1]. Recently, this has stopped being a conceptual future plan and has become a reality. From home appliances to machines in factories, we can easily access IoT devices [2,3,4]. The most distinguishable characteristic of the IoT is a ubiquitous connectivity with real-time response. To achieve this, the device must perform fast processing with low network latency, which depends on the embedded chip performance. Hence, the demand for high-performance chips has continuously increased. To enhance the chip performance, adopting three-dimensional integrated circuits (3D ICs) is inevitable. Because the wire lengths are shorter than those in 2D architectures, 3D ICs have numerous advantages. These include a higher chip performance and bandwidth of circuit functional blocks, less power consumption and noise generation, and a smaller footprint with a higher integration density [5]. Hence, they have attracted the interest of researchers over the last decades [5,6,7,8,9].

However, 3D ICs face several technological challenges that must be addressed. Because the device layers were stacked, the total thickness of the ICs increased. Hence, each layer was thinned. Additionally, 3D ICs require precise alignment between the solder balls and microelectrodes, although this alignment is not easy because of the opaqueness of the Si layers. Furthermore, deformation occurred because each layer exhibited different thermal expansion characteristics. To overcome these challenges, glass substrates have been proposed as an alternative [10,11,12,13,14,15]. In particular, a glass interposer has different advantages which can solve pre-existing challenges. The advantages of this method are as follows: Glass can be thinned using cost-effective processes. Because it is transparent, its alignment is easier than that of a Si interposer. Additionally, the thermal expansion coefficient (CTE) of the glass is adjustable. Thus, it can be matched with the CTE of the device layers to prevent deformation [14]. Hence, glass interposers have been studied extensively over the last decade [10,12,14,16].

There are several methods to generate holes on a glass substrate. These methods include ultrasonic drilling, powder blasting, abrasive jet micromachining (AJM), abrasive slurry jet machining (ASJM), abrasive water-jet machining (AWJM), laser machining, wet etching, deep reactive ion etching (DRIE), plasma etching, spark-assisted chemical engraving (SACE), vibration-assisted micromachining, laser-induced plasma micromachining (LIPMM), and water-assisted micromachining [17]. The aforementioned processes have both advantages and disadvantages. From the mass production point of view, the most important parameter is the shape uniformity between the holes. Additionally, the processing time must also be considered. Selective laser-induced etching (SLE) generates uniform holes with a long processing time. Hence, in this study, we attempted to reduce the processing time to enhance productivity using the SLE technique.

Selective laser-induced etching (SLE) comprises two steps: (1) the local modification of the glass using an ultrashort pulsed laser, and (2) selective chemical etching of the modified area [18]. Based on our previous study, the laser-modified area had an etch rate 333 times faster than that of the unmodified area [19]. Because the physical and chemical properties of the modified area change, they quickly react with the etching chemicals. The altered properties include nanograting generation, volume expansion, and refractive index changes [20,21,22].

In this study, four cases were tested to enhance the etching process speed. These included a single pulse, and double pulses with 213 ps, 10 ns, and 500 ms intervals, respectively. After local modification using an ultrashort pulsed laser under the aforementioned conditions, the modification area was explored using scanning electron microscopy (SEM). Here, we found a variety of nanograting formations in each case and revealed that they affected the etching rate. The etching environment was manipulated to enhance the etching rate. In particular, the temperature of the etching solution was increased to 110 °C. This enhanced the kinetic energy of the solution, thereby making the chemical reaction more active. Based on our study, we can claim that double pulses with ps intervals can enhance the etching rate by more than nanosecond or millisecond intervals.

## 2. Materials and Methods

### 2.1. Substrate Material

We used a borosilicate glass (D 263^®^ T eco, SCHOTT, Mainz, Germany) substrate with a thickness of 0.1 mm (0.1 t) to generate the Through Glass Via holes (TGV). This material uses an eco-friendly refining agent instead of arsenic or antimony. The manufacturer claims that this glass suits radio frequency and high frequency applications, while the IoT devices that we consider in this study will use 5G network technology. Thus, it is a good candidate for a glass interposer base material.

### 2.2. Local Modification with an Ultrashort Pulsed Laser

In this study, we adopted a Bessel beam to enhance productivity. The depth of focus of a Bessel beam is large and can exceed the glass thickness. Hence, a single pulse of a Bessel beam can generate a full-thickness local modification of a glass. A schematic of the experimental setup for the local modification using an ultrashort-pulsed laser with a Bessel beam is shown in Figure 1a. A diode-pumped Yb:KGW ultrashort-pulse laser (Pharos, Vilnius, Lithuania, PH1–20, Light Conversion, center wavelength of 1030 nm) was used as the light source. It can generate burst-mode pulses of 25 pulses with a 213 ps interval and 9 pulses with a 10 ns interval. As the number of pulses increased, the energy of each pulse decreased. Because at least 30 µJ of pulse energy is required to generate local modifications, we could only use double pulses in this study. The pulse repetition rate was set to 100 Hz and synchronized with the motion stage (M-414.2PD, C-863.11, Physik Instruments, Karlsruhe, Germany). The speed of the stage was set to 10 mm/s. Hence, the distance between the local modifications was 100 μm. To investigate the effect of the pulse duration, it is changed from 0.2 to 1 picosecond (ps) with a 0.2 ps increment. As aforementioned, a pulse energy of at least 30 µJ is required for generating local modification in our experimental set up. Thus, for a fair comparison between each case, we set the pulse energy as 100 µJ. Hence, a pulse energy of 200 µJ (100 µJ + 100 µJ) was radiated from the laser. However, a considerable reduction in the pulse energy occurred at the spatial light modulator (LCOS-SLM, X10468-03, Hamamatsu Photonics, Shizuoka, Japan). The pulse energy at the glass surface was measured and calculated using a power meter (Nova II with a 30A-BB-18 sensor; Ophir, Jerusalem, Israel). Only 34% of the pulse energy reached the glass surface. In other words, the irradiated pulse energy on the glass was 68 µJ (34 µJ + 34 µJ). Beam shaping from Gaussian to Bessel involves two steps: first, a Gaussian beam is transformed into a donut beam using the LCOS-SLM. To transform the beam shape, a phase image is applied to the LCOS-SLM. In this study, the phase images were generated using an optical engineering program (VirtualLab Fusion; LightTrans, Jena, Germany) (Figure 1b). In our experimental setup, 64 phase levels were sufficient to generate a donut-shaped beam. Further, a Bessel beam was formed and measured using a beam profiler (FM100-YAG1064-50x; Metrolux, Berlin, Germany). Here, the widths of the Bessel beam were 5.8 μm (FWHM) and length 180 μm (Figure 2). As the glass thickness was 100 μm, 180 μm was enough to generate full thickness local modifications in the glass.

### 2.3. Selective Laser Etching

In this study, SLE was generated using TGV. Because potassium hydroxide (KOH) solution has a high selectivity, it was used as an etchant for glass etching. The initial etching process time was 6 h at 95 °C. As such, this study aimed to reduce the total processing time. Hence, we attempted to increase the etching rate by increasing the etching temperature. Generally, to initiate a chemical reaction, the energy of the molecule must exceed its activation energy. According to the Arrhenius equation, as the temperature increases, the rate constant (frequency of collisions resulting in a reaction) also increases. Thus, an increase in the temperature enhanced the reaction rate. Here, we used 8 mol/L of KOH solution (31% KOH solution), with a boiling temperature of approximately 128 °C (MSDS: P5887, 10% KOH solution: 101 °C, 45% KOH solution: 132 °C). Considering safety, the etching environment was set to 110 °C. Here, the modified glass was dipped in an 8 mol/L KOH solution in a Teflon container. Later, the container was dipped into heat transfer fluid (Therminol D12^®^, Kingsport, TN, USA) in the digital purge control oil bath (WHB-6, Dai Han Scientific, Seoul, Republic of Korea). After etching, the glass samples were rinsed with deionized (DI) water and isopropyl alcohol (IPA). Finally, the cleansing liquid was removed using compressed nitrogen gas.

### 2.4. Analysis Method

In this study, the depths and diameters of the TGV were observed using optical microscopy (OM, KH-8700, Hirox, Tokyo, Japan) and scanning electron microscopy (SEM, S-4800, Hitachi, Tokyo, Japan). To reveal the effect of the pulse interval, side view SEM images were used. To observe the nanograins along the depth direction, we performed additional etching with an HF solution (49% HF, 1:3 DI water) after cracking the glass. The additional etching revealed nanograting on the local modification line.

## 3. Results and Discussion

### 3.1. Pulse Duration Effects on Local Modification and Etching

As aforementioned, a laser pulse with a duration of 0.2 to 1 ps was irradiated on the glass with various conditions (single pulse, double pulses with 213 ps interval, double pulses with 10 ns interval, and double pulses with 500 ms interval). Generally, as the pulse duration became shorter, the intensity became higher. Consequently, when the sample was irradiated with a shorter pulse, the temporal density of the electrons also increased. Thus, we expect that the local modification along the beam propagation direction was affected by the pulse duration. Here, we drew a line with the Bessel beam and etched it at 110 °C using 8 mols/L KOH solution for 5 h. The diameter of the surface was then measured using an optical microscope. To measure the depth, the sample was broken to obtain a cross-section. This was measured using an optical microscope.

Figure 3 shows the effect of pulse duration on the TGV generation.

The result shows that as the pulse duration becomes longer, the diameter and depth of the TGV become smaller and deeper, respectively. In the case of a 0.2 ps pulse duration, more photons were absorbed near the surface. When the intensity was relatively high, it caused more multiphoton absorption. On the other hand, for a pulse duration of 1 ps, the intensity was relatively lower than of the 0.2 ps pulse duration case. Thus, less multiphoton abruption occurs near the surface and more photons can travel deep inside the glass. Hence, a deeper TGV can be generated using a longer pulse duration. In this case, the diameter of the TGV is reduced because of the lower absorption near the surface. In this study, we focused on the sub-picosecond to one-picosecond regimes because of the maximum pulse energy and optical setup efficiency. We claim that a longer pulse duration may generate a deeper TGV.

### 3.2. Pulse Interval Time Effects on Local Modification and Etching

The greatest feature of interest in our study is the difference between the local modifications generated by double pulses at 213 ps and 10 ns intervals. When the laser pulse irradiates the glass, the following phenomena occur sequentially. First, carrier excitation occurs from the time of the photon arrival to a few ps. This includes the absorption of photons and impact of ionization. Further, thermalization occurs simultaneously. This includes carrier–carrier scattering and carrier–phonon scattering. Furthermore, carrier removal occurs. This includes auger recombination, radiative recombination, and carrier diffusion. Finally, the thermal and structural effects were observed. These processes include ablation, evaporation, thermal diffusion, and re-solidification. The time scale of the thermal and structural effects ranges from a few tens of picoseconds to a few microseconds. In particular, re-solidification occurred after a few nanoseconds [23]. In summary, at the beginning, carrier (electron)-related phenomena dominantly occurred. This was followed by the occurrence of the thermal phenomena and structural changes. Thus, in this study, we set the time interval between the two pulses to 213 ps and 10 ns, which represent the electron- and heat-related phenomena regimes, respectively. In the case of a 213 ps interval, additional photons were irradiated during a carrier excitation and thermalization regime. However, in the case of a 10 ns interval, additional photons were irradiated during the carrier removal and thermal/structural effect regimes.

In this study, we hypothesized that adding more energy to the carrier excitation time regime would be more efficient for enhancing photon absorption than adding more energy to the thermalization time regime. Enhancing the kinetic energy of electrons is generally easier than enhancing heat. To test this hypothesis, we performed a local modification using double pulses with the aforementioned time interval. Further, the glass was etched with 110 °C of 8 mol/L KOH solution for 5 h. The diameter of the top surface of the glass was measured after etching. To measure the depth, the glass was broken and cross-sectional images were acquired using an optical microscope. These results confirmed our hypothesis (Figure 4). For every pulse duration case, the 213 ps time interval cases had the deepest TGV holes. In particular, the 1 ps pulse duration case had the deepest TGV hole, with a depth of 22.39 μm.

In addition, the etching depth was increased by increasing the pulse duration for the single, 213 ps interval case, and 500 ms interval cases. However, for 10 ns cases, the etching depth was almost constant. We think that this result supports the aforementioned hypothesis. For the 213 ps interval cases, with an increasing pulse duration, the optical penetration depth of photons also increased because the electrons were in excited states and they were diffused to beam the propagation direction. Meanwhile, for 10 ns cases, most electrons went back to the ground state. However, phonon and heat diffusion occurred. Because the thermal diffusion length was not significantly varied, the etching depth was similar in this case.

To explore the difference between the two cases in detail, we observed the cross section of the glass. Using an optical microscope, we observed the dark line caused by a Bessel beam. However, this was not observed using SEM. Thus, we can conclude that there is a change in density along the local modification line. To verify this, glass was dipped in a diluted HF solution. Then, 10 mL of 12.25% HF solution was added. The glass was then immersed in the solution for 30 s. Figure 5 shows the result. Figure 5a–c show the optical microscopy images after etching with 8 mol/L KOH. In these images, dark lines can be clearly observed. Figure 5d–f show the SEM images after an additional 12.25% HF etching. These nanostructures can be clearly observed. Figure 5a,d are the case of the single Bessel beam. In this case, a periodic nanostructure is formed. The period is 346 ± 14 nm, which is equivalent to the following equation: $p=\frac{\lambda }{2n}$, where p is the period of the nanostructure, λ is the wavelength of the laser (1030 nm), and *n* is the refractive index of the material (1.5123 ± 0.0015). Figure 5b,e show the case of a double pulse with 213 ps intervals. In this case, the nanograting was periodically formed. Its period and thickness of nanograting were 328 ± 27 and 156 ± 22 nm, respectively. (Figure 6) Nanograting was also formed in the case of a double pulse with 10 ns intervals (Figure 5c,f). Its period and thickness were 339 ± 27 and 231 ± 31 nm, respectively. Hence, we can conclude that generating thinner nanograting enhanced the etching rate. Additionally, in the case of a 10 ns interval, we can observe a trace of melting on the nanograting. Hence, they may have experienced the heat. Meanwhile, for a 213 ps interval, we only observed the nanograting. As such, we can claim that adding more energy to the carrier excitation time regime is more efficient for generating TGV.

The aforementioned experiment was performed using 0.1 t borosilicate glass samples. An 8 M KOH solution was prepared and the glass samples were immersed in the solution. After etching for 9 h, the TGV finally formed (Figure 7).

## 4. Conclusions

In summary, we considered the generation and utilization of donut beams using a spatial light modulator (SLM), resulting in the formation of Bessel beams with a broad depth of focus. This study used a selective laser etching technique in which 100 μm thick glass samples were irradiated with a Bessel beam and then etched with 110 °C 8M KOH solution. This study aimed to investigate the effect of electron absorption and phonon absorption in the commercial glass sample. Here, we report experimental results obtained by adjusting the temporal interval between the double pulses. A double pulse with an interval of 213 ps enhanced the kinetic energy of electrons, and a double pulse with an interval of 10 ns enhanced the heat. Our results indicate that the enhancement of the kinetic energy of the electrons was more advantageous in terms of processability than thermal enhancement. This was confirmed through scanning electron microscopy (SEM) images of the local modified zone. We found that narrower nanograting was formed by additional pulse energy after 213 ps. This means that the effect of electron absorption is greater than that of thermal. As expected, the double pulse irradiation with 213 ps interval produced the deepest holes compared to the other conditions. Therefore, a ps interval double pulse is a favorable condition to improve the TGV generation rate.

## Figures and Tables

**Figure 1 micromachines-14-01766-f001:**
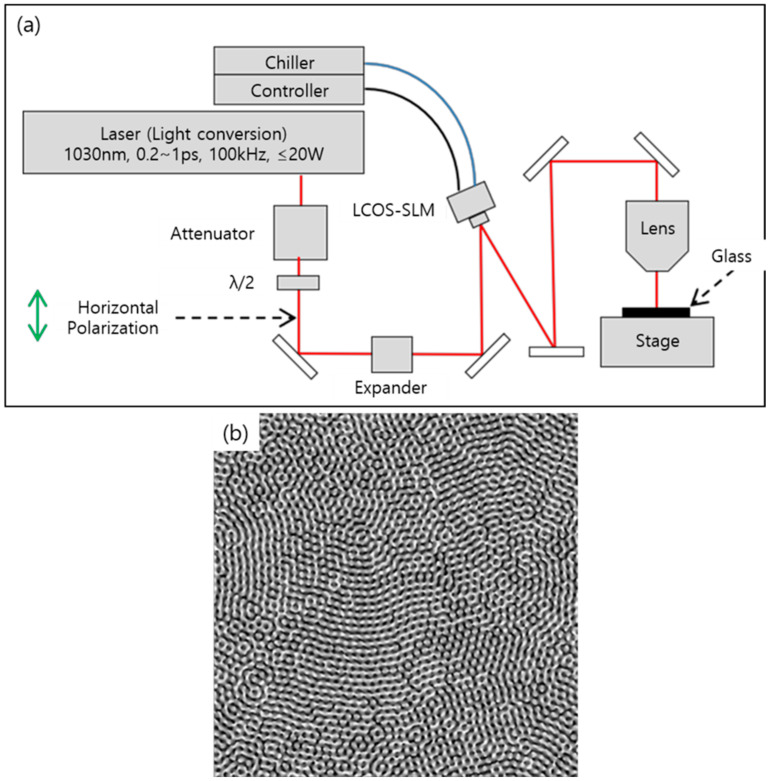
(**a**) Laser systems setup for the Bessel beam generation and local modification and (**b**) digital Fresnel zone plate lens hologram design.

**Figure 2 micromachines-14-01766-f002:**
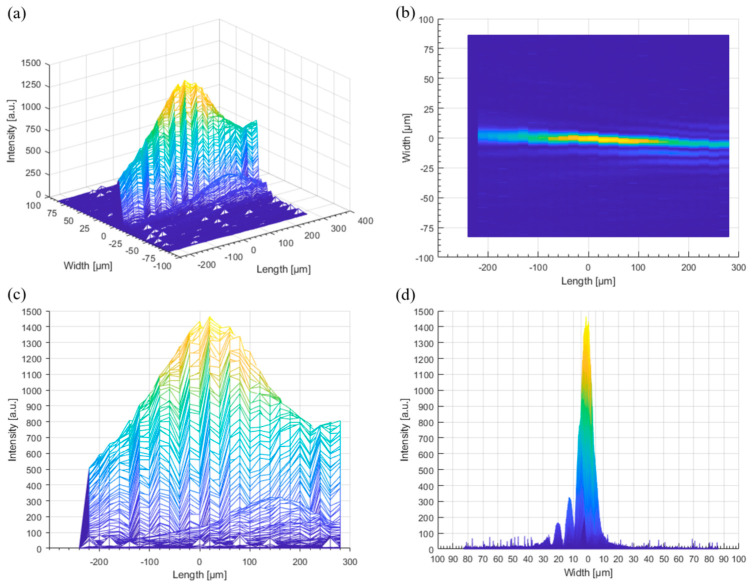
(**a**) Dimetric image: (**b**) top view image, (**c**) side view, and (**d**) front view of the Bessel Beam.

**Figure 3 micromachines-14-01766-f003:**
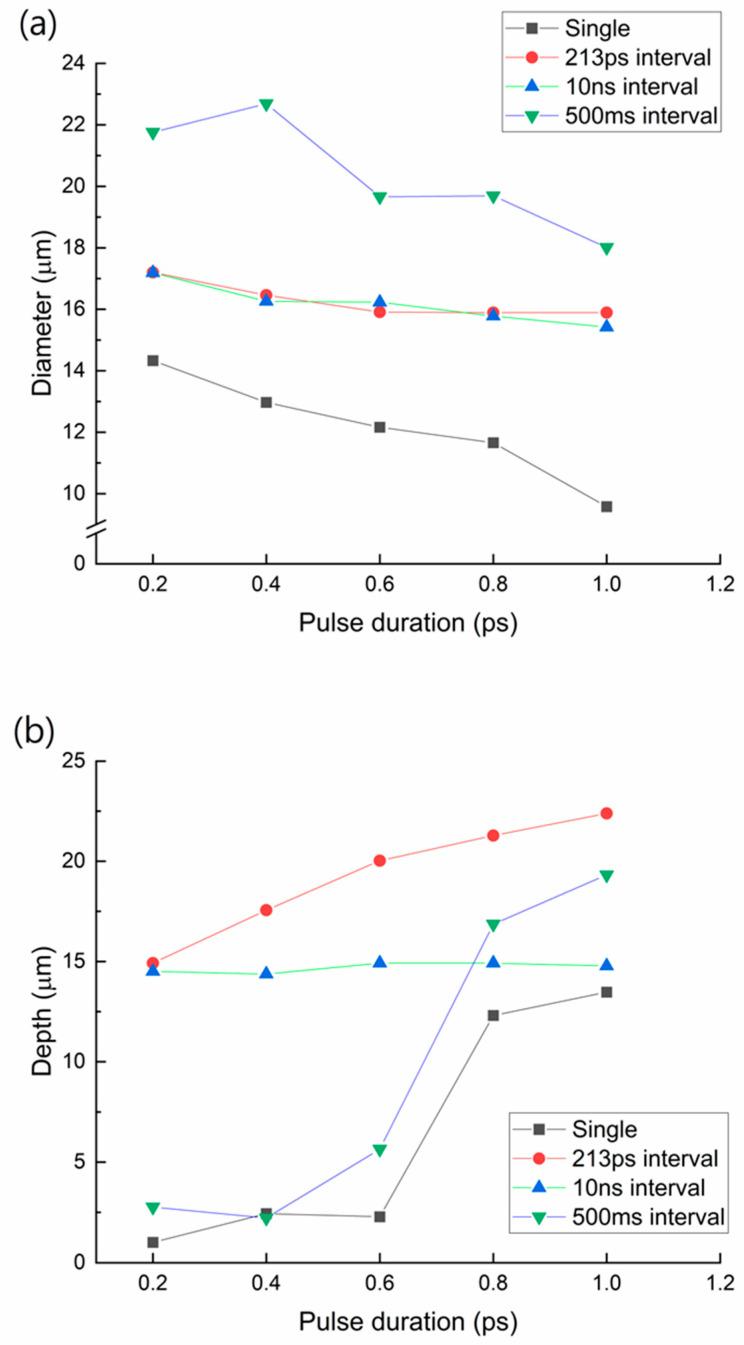
Both the (**a**) diameter and (**b**) depth of the TGV based on the pulse duration.

**Figure 4 micromachines-14-01766-f004:**
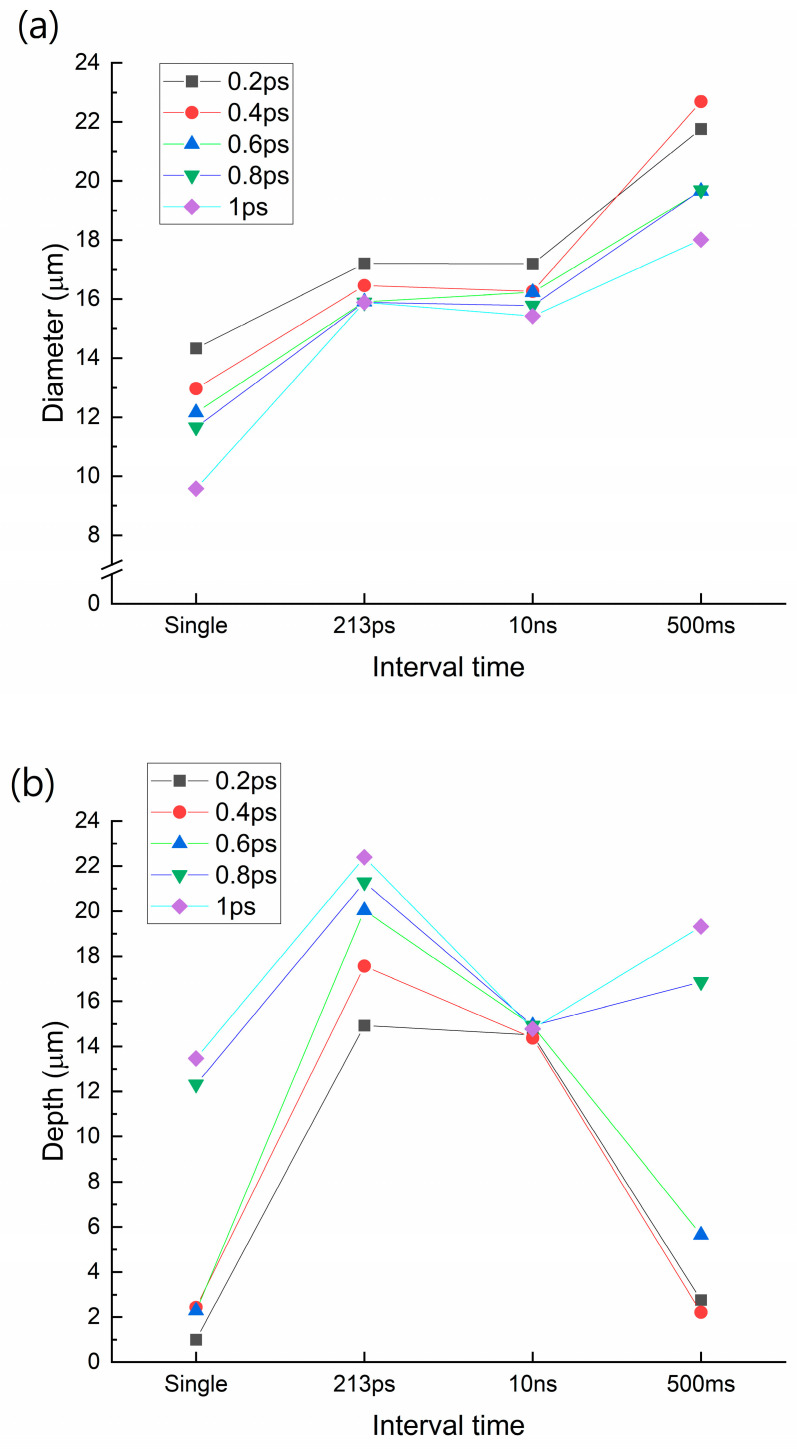
TGV (**a**) diameter and (**b**) depth of each case. This includes the pulse duration from 0.2 to 1 ps and a pulse-to-pulse time interval from a single pulse to the 500 ms interval.

**Figure 5 micromachines-14-01766-f005:**
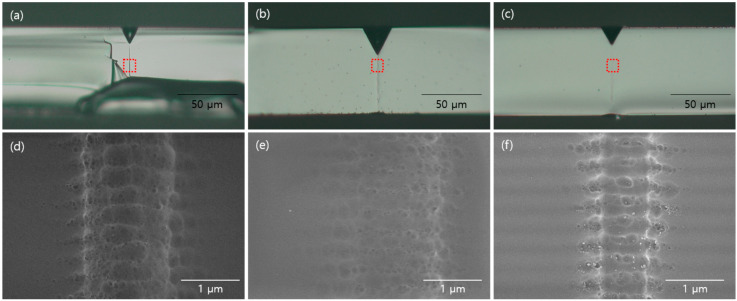
Nano grating formed by the double Bessel beam: (**a**–**c**) optical microscope images and (**d**–**f**) SEM images of each case. Here, pulse duration is fixed as 1 ps. Local modification with (**a**,**d**) single pulse, (**b**,**e**) double pulses with 213 ps interval, and (**c**,**f**) double pulses with 10 ns interval.

**Figure 6 micromachines-14-01766-f006:**
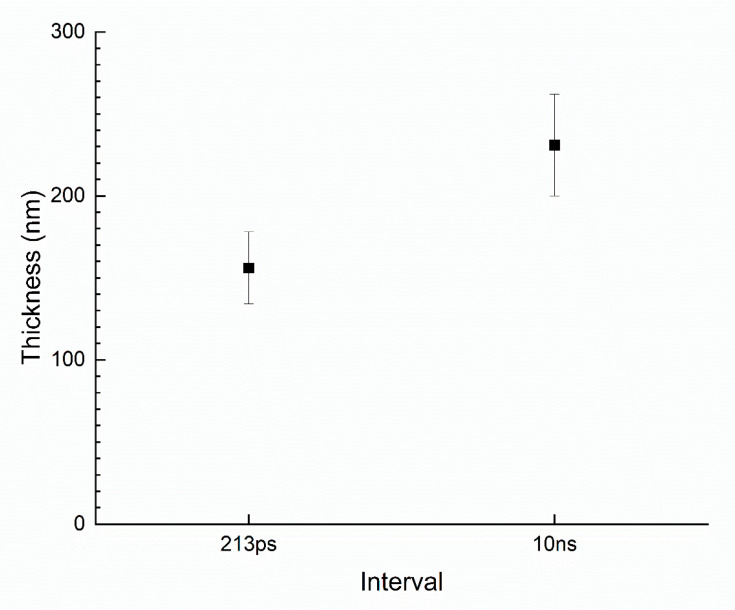
Thickness of nanograting after 12.25% HF etching.

**Figure 7 micromachines-14-01766-f007:**
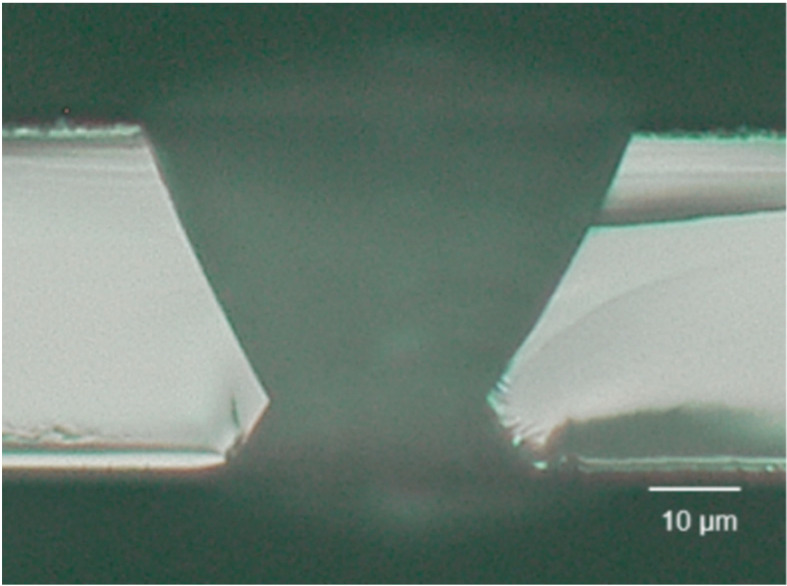
TGV observation of etching for 9 h in KOH 8 M solution.

## Data Availability

Data underlying the results presented in this paper are not publicly available at this time but may be obtained from the authors upon reasonable request.

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
