# Peer review of "Study of Through Glass Via (TGV) Using Bessel Beam, Ultrashort Two-Pulses of Laser and Selective Chemical Etching"

_micromachines, 2023, doi:10.3390/mi14091766_

Round 1

Reviewer 1 Report

The authors studied the through glass via applying Bessel, ultrashort two-pulses of laser, and the selective chemical etching. They have combined the nanosecond laser and picosecond laser with different parameters during the investigation. The results and the arrangement of this manuscript are quite good. I think this paper can be accepted with answering the following questions.

1.        The resolution ration of Fig. 1 is not high enough.

2.        The words in Fig. 3 and Fig. 4 is too small, I think the authors should enlarge them for a better reading.

3.        Considering the mainly research focus of this manuscript, I hope the authors could compare their work with the following references: Dittrich S, Barcikowski S, Gökce B. Plasma and nanoparticle shielding during pulsed laser ablation in liquids cause ablation efficiency decrease. Opto-Electron Adv 4, 200072 (2021).. doi: 10.29026/oea.2021.200072; Dittrich S, Spellauge M, Barcikowski S, Huber HP, Gökce B. Time resolved studies reveal the origin of the unparalleled high efficiency of one nanosecond laser ablation in liquids. Opto-Electron Adv 5, 210053 (2022). doi: 10.29026/oea.2022.210053, and what is the difference if using femtosecond laser (Zhang DS, Li XZ, Fu Y, Yao QH, Li ZG et al. Liquid vortexes and flows induced by femtosecond laser ablation in liquid governing formation of circular and crisscross LIPSS. Opto-Electron Adv 5, 210066 (2022). doi: 10.29026/oea.2022.210066).

4.        Can the authors offer the roughness information about the nanogratings shown in Fig. 5?

The language should be polished before the accepted.

Author Response

  1. We have changed Fig.1
  2. Fig.3 and 4 are also changed
  3. Dittrich et al. is related with plasma shielding when the nanosecond pulse duration laser is irradiated on the surface in liquid atmosphere. However, in this study, we only use ultrashort pulse laser which is less than 1 picosecond. The intervals, so called the time distance, between each pulse are 213 ps and 10 ns. Thus, it is not fair to compare this paper with our study. Zhang et al. is related with liquid vortexes. Again, in this paper, we don’t use any liquid. Thus, I don’t find any reason why we compare our study with this paper as well.
  4. Because a thickness of glass is too small, it is hard to measure the surface roughness of cross section with AFM. Thus, we don’t know an exact value of it. 

Reviewer 2 Report

Dear Authors,

the SLE method is well known in the MEMS community and is already commercially available. It would be good to include references of all leading groups. Averall the background of the process is discribed quite shortly. It should be stated, that this process is already used for production.

 The novelty of the experiments in contrast to other groups is also not well described. 

Regarding the experiments and the results I wonder, if you should investigate some more interval and pulse times. Your assumption regarding the heat related phenomena regimes are good, but should be proofed by experiments. If you choose only 213 ps and 10 ns, this is not a verification of your assumption. The experiment should be a bit more systematic. Especially Fig 4.b creates some questions regarding the different time pulses for 10 ns intervall time. Why does the pulse duration have no influence on the depth of the holes? 

Author Response

We put an additional explain in the manuscript.

"In addition, etching depth is increasing by increasing of pulse duration for single, 213ps interval case, and 500ms interval cases. However, for 10ns cases, the etching depth is almost constant. We think that this result supports aforementioned hypothesis. For 213ps interval cases, as an increasing of pulse duration, optical penetration depth of photon is also increasing. Because the electrons are excited states and they are diffused to beam propagation direction. Meanwhile, for 10ns cases, most electrons go back to ground state. But phonon and heat diffusion occur. Because the thermal diffusion length is not significantly varied, the etching depth is similar in this case."